# A Deep Learning Model with Signal Decomposition and Informer Network for Equipment Vibration Trend Prediction

**DOI:** 10.3390/s23135819

**Published:** 2023-06-22

**Authors:** Huiyun Wang, Maozu Guo, Le Tian

**Affiliations:** School of Electrical and Information Engineering, Beijing University of Civil Engineering and Architecture, Beijing 102616, China; 2108110021030@stu.bucea.edu.cn (H.W.); guomaozu@bucea.edu.cn (M.G.)

**Keywords:** equipment operation trend prediction, variational mode decomposition, ICEEMDAN, quadratic decomposition, informer

## Abstract

Accurate equipment operation trend prediction plays an important role in ensuring the safe operation of equipment and reducing maintenance costs. Therefore, monitoring the equipment vibration and predicting the time series of the vibration trend is one of the effective means to prevent equipment failures. In order to reduce the error of equipment operation trend prediction, this paper proposes a method for equipment operation trend prediction based on a combination of signal decomposition and an Informer prediction model. Aiming at the problem of high noise in vibration signals, which makes it difficult to obtain intrinsic characteristics when directly using raw data for prediction, the original signal is decomposed once using the variational mode decomposition (VMD) algorithm optimized by the improved sparrow search algorithm (ISSA) to obtain the intrinsic mode function (IMF) for different frequencies and calculate the fuzzy entropy. The improved adaptive white noise complete set empirical mode decomposition (ICEEMDAN) is used to decompose the components with the largest fuzzy entropy to obtain a series of intrinsic mode components, fully combining the advantages of the Informer model in processing long time series, and predict equipment operation trend data. Input all subsequences into the Informer model and reconstruct the results to obtain the predicted results. The experimental results indicate that the proposed method can effectively improve the accuracy of equipment operation trend prediction compared to other models.

## 1. Introduction

For public buildings, structural safety is the most important goal in the implementation process of various construction projects. Temporary structures such as work platforms and load-bearing support systems directly affect the construction safety of super high-rise buildings. Scaffolding, as the “heart” of various major building systems, plays an overall role in various major events and construction fields. Once the performance of the scaffold deteriorates it not only seriously affects task execution and production efficiency, but also leads to malignant events, resulting in incalculable losses. Therefore, the safety and stability of temporary frames have attracted much attention. Due to the relatively complex operating environment of the temporary frame, the performance of the parts will gradually age over time, and extreme weather conditions and human factors can also have a certain degree of impact on the parts. Therefore, the equipment is prone to failure, and this failure may be random. However, with the continuous improvement of equipment operation process requirements and increasing maintenance costs, traditional equipment operation trend prediction methods based on timeliness and low accuracy have been unable to meet the needs of effective equipment maintenance and risk avoidance.

Based on the above requirements, researchers conducted research on equipment operation trend prediction. The main prediction of mechanical equipment operating trends models includes mathematical models and deep learning models [1,2,3]. Mathematical models such as wavelet packet decomposition [4], auto-regressive integrated moving average model [5] and empirical mode decomposition [6], show poor predictive ability when dealing with large samples and long time series [7]. Dragomiretskiy et al. [8] proposed a completely non-recursive variational mode decomposition (VMD) method, which is more robust to sampling and noise, and has better results in decomposing complex signals. However, the number of modes can affect the decomposition accuracy. Complete Ensemble Empirical Mode Decomposition with Adaptive Noise (CEEMDAN), proposed by Torres et al. [9], added Gaussian noise to the residual and averaged it multiple times to offset the noise. The decomposed components can be added to obtain the original signal. Colominas et al. [10] proposed ICEEMDAN (Improved complete extensible EMD), which solves the problems of residual noise and pseudo modes in CEEMDAN.

In recent years, the powerful feature extraction ability of deep learning has attracted the attention of researchers, making it widely used in the prediction of mechanical equipment operating trends. Recurrent neural networks can simultaneously model sequential and time dependencies on multiple scales [11]. However, RNN has the same feature extraction ability for all inputs when processing time series. As time increases, RNN will have a problem of decreasing or even disappearing the gradient, resulting in RNN having only short-term memory and not being able to learn well the long-term dependencies of temporal data. The Transformer model was proposed by Vaswani et al. [12], whose core principle is the self-attention mechanism. Compared to the RNN model, the Transformer model exhibits superior performance in capturing remote dependencies. However, Transformer’s self-attention mechanism standardizes the dot product computing method, resulting in a large amount of memory and computing resources required for the computing process, greatly increasing the operating cost [13]. The ProbSparse self-attention mechanism was proposed to improve the Transformer model and achieve good results in predicting long series data [14]. The Informer model solved the limitations of Transformer in processing long time series, greatly reducing the time complexity and memory usage of each layer. Informer uses a generative decoder to obtain long sequence output, requiring only one forward step to output the entire decoded sequence, while avoiding cumulative error propagation during inference.

The main contributions of this paper are summarized as follows:We propose a novel and effective signal decomposition approach for equipment operation trend prediction. In order to solve the problem of difficulty in determining the number of signal decomposition algorithms, an improved sparrow search algorithm optimization decomposition algorithm was proposed.The fuzzy entropy scale of the signal after one decomposition is taken into account by us. A signal decomposition algorithm for VMD-ICEEMDAN quadratic decomposition was constructed by decomposing the component with the highest fuzzy entropy using ICEEMDAN.We innovatively combine the VMD-ICEEMDAN model and the Informer approach, and a novel equipment running trend prediction approach is proposed to improve the prediction results. Through a large number of comparative experiments, it has been proven that the method proposed in this paper is indeed efficient.

## 2. Related Work

It is well known that the trend prediction of equipment operation is a time series prediction problem. Many time series prediction models are based on two strategies:Deep learning models. In Guo et al. [15], a Recurrent Neural Network (RNN) based health indicator was proposed to predict the RUL of bearings. RNN can mine temporal and semantic information from time series. In Qin et al. [16], the root mean square at different times was used as the health indicator, and a new kind of gated recurrent unit (GRU) neural network with dual attention gates was used to predict the trend of achieving health indicators. Still, the error will gradually increase under transient (nonstationary) operating conditions. A novel feature-attention-based end-to-end approach was proposed for RUL prediction by Liu et al. [17]. Convolutional neural networks (CNN) are applied to capture local features from the output sequences of BGRU, but the prediction performance needs to be improved. Yang, et al. [18] introduced Informer into the time series forecasting of motor bearing vibration, which reduced the error accumulation in forecasting.The combination of signal decomposition and deep learning models. Nowadays, very few studies use only one of the linear, nonlinear, wavelet transform, or other mathematical models for trend prediction of equipment. The majority of research incorporates machine learning or deep learning models. In Wang et al. [19], a novel method of rubbing fault diagnosis based on variational mode decomposition (VMD) was proposed, which could non-recursively decompose a multi-component signal into a number of quasi-orthogonal intrinsic mode functions. The fault separation upon the principles of empirical mode decomposition (EMD), envelope analysis and the pseudo-fault signal was built by Singh, and Zhao [20], which solved the mode mixing problem inherent in EMD. Qiu et al. [21] used EMD to decompose the power load series to obtain a series of intrinsic mode functions and then used deep belief networks to predict the intrinsic mode functions. Liang et al. [22] used ICEEMDAN to decompose the original series into different levels of components and then used a network composed of short-term and short-term memory, convolutional neural networks, and convolutional attention modules (LSTM-CNN-CBAM) to predict all components, proving that the decomposed time series components are superior to the undissociated time series components in terms of model prediction accuracy.

At present, most research does not consider both the impact of the volatility of time series and the limitations of prediction models, so it cannot be applied well to the machinery industry. Above all, the trend prediction model of equipment based on Signal decomposition and Informer is proposed in this paper. The combined model is composed of Variational Mode Decomposition (VMD) and Improved adaptive white noise complete set empirical mode decomposition (ICEEMDAN) and Informer. VMD is used for decomposing the complex original sequence into a series of low-complexity and different-frequency subsequences. Moreover, the fuzzy entropy of the subsequence is calculated. ICEEMDAN is applied in decomposing the subsequence with the largest fuzzy entropy named IMF_M to generate a series of new intrinsic mode functions. The cross-correlation coefficient of the new intrinsic mode functions and the IMF_M are calculated to eliminate irrelevant components, and all the remaining components are utilized to train and test the Informer model for trend prediction. The obtained results are reconstructed to obtain the final prediction result.

## 3. Methodology

### 3.1. VMD and Its Improvement

Variational mode decomposition (VMD) extracts the signal by using the idea of solving a variational problem to decompose an original signal into multiple signals with different center frequencies and without losing the characteristics of the original signal so that the effective separation of the frequency domain part and components of the signal can be achieved adaptively.

The VMD algorithm to decompose the signal can be understood as the construction and solution of the variational problem. VMD calculation equations are shown in Equations (1)–(6) [23,24]:(1)$$\left\{\begin{array}{c} \underset{\left\{\nu_{k}\right\},\left\{\omega_{k}\right\}}{\min}\left\{\sum_{k=1}^{K}{\left\|\partial_{t}\left[\left(\delta_{t}+\frac{j}{\pi t}\right)*\nu_{k}\left(t\right)\right]e^{-jw_{k}t}\right\|}_{2}^{2}\right\}\\ s.t.\sum_{k=1}^{K}\nu_{k}=s(t) \end{array}\right.$$
(2)$$L\left(\left\{v_{k}\right\},\left\{\omega_{k}\right\},\tau \right):=\alpha {\sum_{k}\left\|\partial_{t}\left[\left(\delta \left(t\right)+\frac{j}{\pi t}\right)*v_{k}\left(t\right)\right]e^{-j\omega_{k}t}\right\|}^{2}+{\left\|s(t)-\sum_{k}v_{k}\left(t\right)\right\|}^{2}+\langle \tau \left(t\right),s\left(t\right)-\sum_{k}v_{k}\left(t\right)\rangle$$
(3)υ⌢kn+1ω=s⌢ω−∑i≠kυ⌢iω+τ⌢ω21+2αω−ωk2
(4)$$\omega_{k}^{n+1}=\frac{\int_{0}^{\infty }\omega {\left|\upsilon_{k}^{n+1}\left(\omega \right)\right|}^{2}d_{\omega }}{\int_{0}^{\infty }{\left|\upsilon_{k}^{n+1}\left(\omega \right)\right|}^{2}d_{\omega }}$$
(5)$${\overset{⌢}{\tau }}^{n+1}\left(\omega \right)={\overset{⌢}{\tau }}^{n}\left(\omega \right)+\tau \left(\overset{⌢}{s}\left(\omega \right)-\sum_{k}{\overset{⌢}{\upsilon }}_{k}^{n+1}\left(\omega \right)\right)$$
(6)$$\sum_{k}{\left\|{\overset{⌢}{\upsilon }}_{k}^{n+1}-{\overset{⌢}{\upsilon }}_{k}^{n}\right\|}_{2}^{2}/{\left\|{\overset{⌢}{\upsilon }}_{k}^{n}\right\|}_{2}^{2}<\varepsilon$$

Equations (1)–(6) are the equations for decomposing the equipment operating data of traditional VMD, where *s*(*t*) is the original component, *K* is the number of decompositions, $\left\{v_{k}\right\}=\left\{v_{1},\cdots ,v_{k}\right\}$ represents the decomposed IMF component, $\left\{w_{k}\right\}=\left\{w_{1},\cdots ,w_{k}\right\}$ represents the central frequency of each component, α represents the penalty factor, $\tau \left(t\right)$ is the Lagrange multiplier operator, ω is the Frequency, $\overset{⌢}{s}\left(\omega \right)$ is the Fourier transform of the signal *s*(*t*), ${\overset{⌢}{\upsilon }}_{k}^{n+1}\left(\omega \right),\;\overset{⌢}{s}\left(\omega \right)$ and $\overset{⌢}{\tau }\left(\omega \right)$ are the Fourier transforms to ${\overset{⌢}{v}}_{k}^{n}\left(t\right)$, s(t)**,**
τ(t), respectively.

The VMD decomposition process is the process of solving this variational problem, i.e., solving the modal function and the central frequency when the sum of the estimated bandwidths of the central frequencies of the components is minimized.

Assuming that the multi-component signal is decomposed into K modal components of finite bandwidth and the constraint is that the modal sum is equal to the input signal, the constrained variational expression is constructed as shown in Equation (1). The complete algorithm for VMD is summarized in Algorithm 1.
**Algorithm 1: VMD**Initialize v⌢k1, w⌢k1, τ⌢k1, n→0**Repeat**n→n+1     **For**
k=1:K
**do****Update** v⌢k for all w≥0:   υ⌢kn+1ω−>s⌢ω−∑i≠kυ⌢iω+τ⌢ω21+2αω−ωk2
**Update**
$\;w_{k}$:   $\omega_{k}^{n+1}->\frac{\int_{0}^{\infty }\omega {\left|\upsilon_{k}^{n+1}\left(\omega \right)\right|}^{2}d_{\omega }}{\int_{0}^{\infty }{\left|\upsilon_{k}^{n+1}\left(\omega \right)\right|}^{2}d_{\omega }}$
 **end**
**for**
Dual ascent for all w≥0: ${\overset{⌢}{\tau }}^{n+1}\left(\omega \right)->{\overset{⌢}{\tau }}^{n}\left(\omega \right)+\tau \left(\overset{⌢}{s}\left(\omega \right)-\sum_{k}{\overset{⌢}{\upsilon }}_{k}^{n+1}\left(\omega \right)\right)$
**until** convergence: $\sum_{k}{\left\|{\overset{⌢}{\upsilon }}_{k}^{n+1}-{\overset{⌢}{\upsilon }}_{k}^{n}\right\|}_{2}^{2}/{\left\|{\overset{⌢}{\upsilon }}_{k}^{n}\right\|}_{2}^{2}<\varepsilon$


In the actual equipment operation scenario, the parameters of VMD must be determined, such as the number of decompositions and penalty factor. If the set *K* is less than the number of useful components in the signal to be decomposed, it leads to modal aliasing; if the set *K* is greater than the number of useful components in the signal to be decomposed, it leads to the generation of some useless spurious components. Therefore, the determination of the *K* value is very important for VMD. The penalty factor α determines the bandwidth of the IMF component. The smaller the penalty factor, the larger the bandwidth of each IMF component, making some components contain other component signals; the larger the value of α, the smaller the bandwidth of each IMF component, making some signals lost in the decomposed signal. These problems make it difficult to determine the parameters of the decomposition algorithm.

In Zhang et al. [25], the parameter-adaptive VMD method based on the grasshopper optimization algorithm (GOA) was proposed to analyze vibration signals from rotating machinery. The global optimization performance of GOA was demonstrated by Shahrzad et al. [26] and had been used for the parameter optimization of various algorithms because of its being simple and efficient. However, the researchers found that the population diversity and convergence accuracy of the algorithm need to be increased and improved. In Qin et al. [27], the sparrow search algorithm and the VMD method are combined to decompose the original data, so that the decomposition results are more rapid and accurate. However, the performance of the sparrow search algorithm will degrade rapidly when the optimal solution of the problem to be solved is far from the origin.

In summary, this paper proposes an improved sparrow search algorithm (ISSA) to optimize the VMD algorithm. The basic sparrow algorithm uses a random initialization method to generate the initial population when solving the optimal solution problem, which may produce the problem of uneven distribution of the initial sparrow population, resulting in poor diversity of the sparrow population. In order to enrich the diversity of the sparrow population, the Fuch chaos mapping model is introduced to generate a diverse chaotic initialized sparrow population by using the characteristics of Fuch chaos mapping which has a better iteration speed and produces a uniformly distributed chaotic sequence between [0, 1]. Fuch chaotic mapping expression as Equation (7). With a limited number of iterations, the producer’s moving range will gradually shrink, making the algorithm’s global search ability decrease. To address this problem, this paper introduces the positive cosine algorithm to dynamically adjust the producer’s location formula to balance the two processes of global expansion and local optimization, so that SSA has stronger global exploitation capability in the early stage and stronger small-range search capability in the later stage. The improved explorer position update formula is shown in Equation (8), where t represents the number of current iterations, and $X_{i,j}$ denotes the position information of the ith sparrow in the jth dimension. $R_{2}\left(R_{2}\in \left(0,1\right]\right)$ and $ST\left(ST\in \left(0.5,1\right]\right)$ denote the warning value and the safety value, respectively. In the process of finding the optimal solution, Levy flight can not only perform a local search in short distances but also a global search in long distances. Therefore, Levy flight can serve to enhance the local search ability when searching near the optimum, effectively solving the problem that standard SSA may fall into local optimum. The improved follower position equation and Levy flight step equation are shown in Equations (9) and (10), where $X_{P}$ is the optimal position currently occupied by the discoverer and $X_{worst}$ indicates the current global worst position. A denotes a 1 × d matrix where each element is randomly assigned a value of 1 or −1 and $A^{+}=A^{T}{\left(AA^{T}\right)}^{-1}$. The specific algorithm flow is shown in Figure 1.
(7)$$X_{n+1}=\cos\left(1/X_{n}^{2}\right)$$
(8)$$X_{i,j}^{t+1}=\left\{\begin{array}{ll} X_{i,j}^{t}+r_{1}\cdot \sin\left(r_{2}\right)\times \left|r_{3}\cdot x_{best}^{t}-X_{i,j}^{t}\right| & ifR_{2}<ST\\ X_{i,j}^{t}+r_{1}\cdot \cos\left(r_{2}\right)\times \left|r_{3}\cdot x_{best}^{t}-X_{i,j}^{t}\right| & ifR_{2}\geq ST \end{array}\right.$$
(9)$$X_{i,j}^{t+1}=\left\{\begin{array}{ll} Q\cdot \exp\left(\frac{X_{worst}^{t}-X_{i,j}^{t}}{i^{2}}\right) & if\;i>n/2\\ X_{P}^{t+1}+s\cdot \left|X_{i,j}^{t}-X_{P}^{t+1}\right|\cdot A^{+}\cdot L & otherwise \end{array}\right.$$
(10)$$s=\mu /{\left|\beta \right|}^{1/\beta },\;0<\beta \leq 2$$

### 3.2. ICEEMDAN

Improved adaptive white noise complete set empirical mode decomposition (ICEEMDAN) is developed from the adaptive noise complete ensemble empirical mode decomposition (CEEMDAN). Unlike CEEMDAN which adds Gaussian white noise directly during the decomposition process, ICEEMDAN adds special noise $E_{k}\left(w^{\left(i\right)}\right)$ when extracting the kth layer IMF. That is, the kth layer IMF obtained after the Gaussian white noise is decomposed by EMD, and to obtain a unique residual, the IMF obtained by ICEEMDAN decomposition is the difference between the existing residual signal and its local mean. The ICEEMDAN method solves the problem of residual noise and pseudo-modalities in CEEMDAN by reducing the useless modal components [10]. The ICEEMDAN’s algorithm is as follows:Add a set of white noise $w^{\left(i\right)}$ to the original sequence, constructing a sequence $x^{\left(i\right)}=x+x^{\left(i\right)}=x+\beta_{0}E\left(w^{\left(i\right)}\right)$, The first set of residuals is obtained: $R_{1}=\left(N(x^{\left(i\right)})\right)$.Calculation of the first modal component: $d_{1}=x-R_{1}$.Continue to add white noise and use local mean decomposition to calculate the second set of residuals $R_{1}+\beta_{1}E\left(w^{\left(i\right)}\right)$. Define the second modal component d_2_.
$$d_{2}=R_{1}-R_{2}=R_{1}-\left(N\left(R_{1}+\beta_{1}E\left(\omega^{\left(i\right)}\right)\right)\right)$$Calculate the *K*th residual $R_{k}=\left(N\left(R_{k-1}+\beta_{k-1}E\left(w^{\left(i\right)}\right)\right)\right)$ and modal component $d_{k}=R_{k-1}-R_{k}$.Until the end of the computational decomposition, all modalities and residual numbers are obtained.

Where *x* is the signal to be decomposed, $E_{k}\left(\cdot \right)$ denotes the kth order modal component generated by the EMD decomposition, $N\left(\cdot \right)$ denotes the local mean of the generated signal, and $w^{\left(i\right)}$ represents the Gaussian white noise.

### 3.3. Informer

Transformer clearly has superior performance than RNN in capturing long-term dependencies. However, Transformer suffers from high secondary computational complexity of the self-attention mechanism, memory bottleneck of stacked layers under a long sequence of inputs, and slow inference when predicting long outputs. The ProbSparse self-attention mechanism of the Informer model achieves $O\left(L\log L\right)$ for time complexity and memory usage, which solves the Transformer’s problem of high computational complexity. In addition, Informer’s autocorrelation distillation operation highlights features with high attention scores on J stacking layers and greatly reduces the spatial complexity. Therefore, the model can receive long sequence inputs. Informer’s generative decoder, which directly predicts in one time and multiple steps, avoids the accumulation of errors generated by single-step prediction, improves the prediction accuracy and reduces the prediction time. The unit structure of Informer is shown in Figure 2, where the left side represents the encoder, which consists of the ProbSparse self-attention block (PSB) and distillation stack. The ProbSparse self-attention block (PSB) halves the input for each layer of convolution and pooling (C&P) operation. The right side represents the decoder, which receives long sequence inputs and interacts with the encoded features through multi-head attention, and finally predicts the output target part directly in one go.

The classical Transformer model calculates self-attention based on the input triplet (query, key, value), which performs the scaled dot-product as $A(Q,K,V)=Soft\max(QK^{T}/\sqrt{d})V$, where $Q\in R^{L_{Q}\times d},K\in R^{L_{K}\times d},V\in R^{L_{V}\times d}$, and *d* is the input dimension. Let $q_{i},k_{i},v_{i}$ stand for the ***i***th row in *Q, K, V,* respectively. The formula for calculating the weighted value of the ***i***th query is shown in Equation (11):(11)$$A\left(q_{i},K,V\right)=\sum_{j}\frac{k\left(q_{i},k_{i}\right)}{\sum_{l}k\left(q_{i},k_{l}\right)}\upsilon_{j}={Ε}_{p\left(q_{i},k_{l}\right)}\left[V_{j}\right]$$
where $p\left(k_{j}|q_{i}\right)=k\left(q_{i}|k_{j}/\sum_{l}k\left(q_{i}|k_{j}\right)\right)$ and $k\left(q_{i}|k_{j}\right)$ selects the asymmetric exponential kernel $\exp(q_{i}K_{j}^{T}/\sqrt{d})$.

Since there are two ∑ in the computation, self-attention requires $O(L_{Q}L_{K})$ memory as well as the computation of quadratic dot product as a cost. It is shown that the distribution of self-attention Probability is sparse, and only a few dot product calculations of *Q* and *K* dominate the distribution after softmax [14]. The ProbSparse Self-attention mechanism in informer uses KL scatter to define the sparsity of the ***i***th query metric and greatly reduces the time complexity and memory usage. The sparsity evaluation formula is shown in Equation (12), where the first term of $q_{i}$ is the Log-Sum-Exp(LSE) over all keys, and the second term is their arithmetic mean.
(12)$$M\left(q_{i},K\right)=\ln\sum_{j=1}^{L_{K}}\exp\left(\frac{q_{i}k_{j}^{T}}{\sqrt{d}}\right)-\frac{1}{L_{K}}\sum_{j=1}^{L_{K}}\frac{q_{i}k_{j}^{T}}{\sqrt{d}}$$

Based on the above theory, the formula to achieve ProbSparse Self-attention is obtained as shown in Equation (13):(13)$$A\left(Q,K,V\right)=soft\max\left(\frac{\bar{Q}K^{T}}{\sqrt{d}}\right)V$$
where $\bar{Q}$ is a sparse matrix of the same size of *q*.

### 3.4. Deep VMD-ICEEMDAN-Informer Prediction Model

Equipment operation testing data are characterized by instability and random noise, which is caused by the complex environment in which the equipment is located. Equipment operation time, weather, temperature, human and other factors will have an impact on the health status of the equipment, and the degree of influence of each factor on the equipment operation trend may not be visually reflected in the test data. For example, in extreme bad weather, due to enhanced wind, the equipment will inevitably generate more substantial vibration information than usual, while factors such as temperature, humidity, and light are difficult to directly link to equipment vibration information. Therefore, the accurate extraction of information inside the equipment operation data plays an important role in carrying out prediction tasks in a rational and efficient manner.

To be able to extract more information characterizing the equipment operation trend, this paper uses VMD to decompose the original operation data into several VMFs, which have more significant regularity and more obvious signal characteristics compared with the original signal. Previous studies have directly modeled the components after VMD decomposition without considering the influence that the decomposed complex components still have on the prediction results. The magnitude of fuzzy entropy represents the complexity of the signal, i.e., the complexity of the feature information contained inside the signal. The fuzzy entropy of VMFs is calculated to determine whether the components decomposed by VMD still contain too many internal features [28]. The fuzzy entropy is calculated by Equation (14):(14)$$FuzzyEn\left(n,m,r,N\right)=\ln O^{n}\left(m,r\right)-\ln O^{n+1}\left(m,r\right)$$

The ICEEMDAN decomposition technique is used to obtain a series of intrinsic modal components (IMF) by quadratic decomposition of the overly complex VMF_M to improve the prediction accuracy of the model for complex signals, and then improve the prediction accuracy of the model as a whole. The correlation coefficient can describe the degree of correlation between IMF components and VMF_M. The formula for calculating the correlation coefficient is shown in Equation (15), where: $r_{XY}$ is the number of interrelationships between sequences X and Y; $\bar{X}$ and $\bar{Y}$ are the average value of X and Y; *N* is the number of sequences.
(15)$$r_{XY}=\frac{\sum_{i=1}^{N}\left(X_{i}-\bar{X}\right)\left(Y_{i}-\bar{Y}\right)}{\sqrt{\sum_{i=1}^{N}{\left(X_{i}-\bar{X}\right)}^{2}}\sqrt{\sum_{i=1}^{N}{\left(Y_{i}-\bar{Y}\right)}^{2}}}$$

The IMF components decomposed by ICEEMDAN may contain spurious components. The correlation coefficients between the normalized IMF_M autocorrelation function and the auto-correlation function of each order IMF component are solved, and the correlation coefficients are compared with the correlation coefficient threshold to select the IMF components with a strong correlation with VMF_M from the quadratic decomposition IMF components and eliminate the redundant spurious components, and thus can improve the accuracy of subsequent signal feature extraction. Considering that the traditional RNN model may produce gradient disappearance and explosion when dealing with long sequence data, the Informer model with high operational accuracy and high speed is used to predict each component.

The proposed time prediction method of equipment operation trend (Figure 3) is divided mainly into three modules, i.e., the data collection module, decomposition module, and prediction module.

For the problems of instability and large random noise of equipment operation monitoring data, a prediction model of equipment operation trend based on ISSA-VMD-ICEEMDAN and Informer model, called VIInformer, is constructed, and its flowchart is shown in Figure 4. The specific steps are as follows:Step 1The number of decompositions K and the penalty factor α are obtained according to the ISSA optimized VMD algorithm proposed in Section 3.1. Decomposition of equipment operation process monitoring data into K subseries VMFs using the VMD decomposition algorithm.Step 2Calculate the fuzzy entropy of the VMF of the subseries decomposed in Step 1. The magnitude of the fuzzy entropy reflects the complexity of the signal; the higher the complexity of the signal, the larger the fuzzy entropy value.Step 3The subseries VMF_M with the highest fuzzy entropy is subjected to ICEEMDAN quadratic decomposition; the modal components obtained by quadratic decomposition often contain some pseudo-modal components, which are irrelevant and do not reflect the characteristics of IMF_M. Therefore, the pseudo-modal components need to be eliminated, and the number of interrelationships of the components from the ICEEMDAN secondary decomposition is calculated to eliminate the irrelevant components. Then the components are denoised in order to avoid the data failing to converge and the model training time being too long; the components are normalized.Step 4Informer prediction models are built and predicted for the subsequences generated by Step 2 and Step 3, respectively.Step 5The individual subsequence prediction results obtained from Step 4 are reconstructed to obtain the final results of the equipment operation trend prediction.

## 4. Experiments and Analysis

### 4.1. Experimental Data

The paper uses real-time temporary equipment operation monitoring data from 22 November 2021 to 14 March 2022 for the Beijing 2022 Winter Olympic Games in Yanqing for arithmetic analysis. The data collection system of this competition area collects data 2–3 times per second and has collected more than 300,000 data. The types of sensors used to collect data on the operation of the equipment are inclination sensors and vibration sensors. The sensors transmit the message to the system’s equipment operation safety monitoring platform through a controller area network (CAN) bus and Ethernet and save the data in a time series database. To verify the various indicators of the sensor, we conducted equilibrium tests on the inclination sensor, vibration table tests on the vibration sensor, and long-term low-temperature box stability tests. The test experiment diagram is shown in Figure 5.

However, there may be false positives, missing positives, and delays in the data transmission. Therefore, missing values and outliers in the equipment operation dataset need to be identified and processed. In this study, the same mean interpolation method is used for processing the outlier and missing value of the original data.

Due to the high volatility and randomness of the original sequence, direct prediction of the original sequence will appear to cause large errors. In order to reduce the prediction error, a signal decomposition algorithm is introduced to consider the signal decomposition of the original sequence and extract the feature information inside the sequence for prediction.

The min-max normalization method is applied to equipment operation data in this experiment [29].

### 4.2. Setup of the Parameters and Comparison

#### 4.2.1. Parameters

The following parameters were used for the ISSA-VMD algorithm: In fact, a small number of iterations can result in the algorithm being unable to find the global optimal solution. However, an excessive number of iterations can also waste computing resources. In general, the range of iterations is selected between 50 and 500 [30]. In this paper, in order to find the optimal solution and control the experimental time, we chose a maximum number of iterations of 100. The number of variables to be optimized is set to 2, that is, the number of VMD decompositions K and the penalty factor α, the maximum number of iterations to 100. A small population size can cause the algorithm to fall into local optima, while a large population size can reduce algorithm efficiency. The population size is related to the scale of the problem. The population size is 5 to 10 times the size of the problem [30]. Since the number of variables to be optimized in this paper is 2, the population size was set at 20, the upper limits of K and α are 10 and 2500, respectively, and the lower limits are 2 and 100, and the convergence tolerance is set to 10^−7^. The following parameters were used for the ICEEMDAN algorithm: the noise standard deviation is set to 0.2 according to experience, the number of realizations is set to 50, the maximum number of sifting iterations allowed is set to 50. The parameters of Informer are set as shown in Table 1.

#### 4.2.2. Comparison

In order to verify the accuracy of the algorithm in this paper, the original signal is selected and input directly into Informer, VMD−Informer, ICEEMDAN−Informer, and VIInformer, for experimental verification. The purpose of this experiment is to verify whether the prediction accuracy of the model in this paper is improved compared with the single model and the simple combined model. In addition, this paper conducts lateral experiments on the model. The classic deep learning models LSTM, GRU, and CNN are selected as benchmark models. In order to further verify the accuracy of the model, we combine three classic deep learning models LSTM, GRU, and CNN with signal decomposition algorithms, respectively. The purpose of this experiment is to verify the fit between the decomposition algorithm and different depth models.

The prediction algorithm in this study uses the mean square error (MSE), mean absolute error (MAE), and root mean square error (RMSE) as evaluation indexes in the experiments [14], as shown in the following Equations (16)–(18). For ***n*** samples, define the true value as $y=\left\{y_{1},\cdots ,y_{n}\right\}$ and the predicted value as $\bar{y}=\left\{{\bar{y}}_{1},\cdots ,{\bar{y}}_{n}\right\}$. Each evaluation indicator represents the deviation of the predicted value from the true value, and a smaller value means a better model effect.
(16)$$MSE=\frac{1}{n}\sum_{i=1}^{n}{\left({\bar{y}}_{i}-y_{i}\right)}^{2}$$
(17)$$MAE=\frac{1}{n}\sum_{i=1}^{n}\left|{\bar{y}}_{i}-y_{i}\right|$$
(18)$$RMSE=\sqrt{\frac{1}{n}\sum_{i=1}^{n}{\left({\bar{y}}_{i}-y_{i}\right)}^{2}}$$

### 4.3. Experimental Results and Analysis

Four data recorded by inclination and vibration sensors were used in this study. They are derived from the real world, which makes the experimental results more applicable.

#### 4.3.1. *X*-Axis Vibration Data from Inclination Sensor (Data 1)

Data 1 are the vibration data of the *X*-axis recorded by the inclination sensor. Figure 6a shows the results of the ISSA-VMD decomposition of the trend series. Figure 6b displays the results of calculating the fuzzy entropy for the components after ISSA−VMD decomposition. It can be seen that the fuzzy entropy of IMF_2 is the largest, so the ICEEMDAN secondary decomposition is used for IMF_2. Figure 6c shows the results of ICEEMDAN decomposition for the IMF_2. Figure 7 shows the prediction results with the VIInformer of Data 1. Table 2 shows the results of comparative trials.

As shown in Table 2, by performing ablation experiments, the accuracy of the Informer prediction model based on VMD decomposition is improved compared with the original model, the accuracy of the Informer prediction model based on ICEEMDAN decomposition is reduced, and the prediction accuracy of the Informer model based on VMD−ICEEMDAN secondary decomposition is higher than that of the rest of the models.

To verify the prediction effectiveness of the adopted informer prediction model, a cross-sectional comparison experiment is used. As can be seen from Table 2, by comparing the prediction errors with LSTM, GRU, and CNN, respectively, it can be seen that the Informer model performs best on the VMD−ICEEMDAN quadratic decomposition method, and the model mean square error (MSE) is 0.1792, 0.1739, and 0.2637 lower than LSTM, GRU, and CNN, respectively. The mean square error (MSE) of VIInformer is the smallest. It can be observed that using VMD decomposition is beneficial for predicting results. However, ICEEMDAN decomposition has a negative impact on the predicted results. Through comparison with LSTM, GRU and CNN, we have reached preliminary conclusions that Informer has the best fit with the decomposition model.

#### 4.3.2. *Y*-Axis Vibration Data from Inclination Sensor (Data 2)

Data 2 are the equipment *Y*-axis vibration data recorded by the inclination sensor. Figure 8a shows the result of ISSA−VMD decomposition for Data 2. After calculating the fuzzy entropy of the components after ISSA−VMD decomposition, it was found that the component with the highest fuzzy entropy is IMF_4, which is shown in Figure 8b. The ICEEMDAN quadratic decomposition of IMF_4 yields the results shown in Figure 8c, which shows the results of ICEEMDAN decomposition for the IMF_4. Figure 9 shows the prediction result of Data 2 in our method. We also conduct comparative experiments on Data 2, which is shown in Table 3.

In Table 3, compared with other methods, VIInformer performs best in the face of data mutation. The values of MAE are 0.0992 for Informer, 0.0879 for VMD−Informer, 0.1368 for ICEEMDAN-Informer, 0.0984 for VMD−ICEEMDAN−LSTM, 0.0897 for VMD−ICEEMDAN−GRU, 0.1318 for VMD−ICEEMDAN−CNN and 0.0655 for VIInformer. Accordingly, compared to other methods, the method in this paper improves 0.0337, 0.0224, 0.0713, 0.0329, 0.0242 and 0.0663, respectively. The forecasting errors show that Informer with ICEEMEDAN and CNN with VMD−ICEEMDAN have poor forecasting results, while Informer with VMD and VIInformer are able to achieve good prediction results. The reason is that VMD decomposition can indeed reduce the complexity of the original sequence. VMD−ICEEMDAN enables decomposition-prediction models to have better feature extraction and generalization capabilities. False components generated by ICEEMDAN decomposition reduce the feature extraction ability of the prediction model. The weak feature extraction ability of CNN leads to the poor prediction performance of the model.

#### 4.3.3. *X*-Axis Vibration Data from Vibration Sensor (Data 3)

Data 3 are the equipment *X*-axis vibration data recorded by the vibration sensor. Figure 10a shows the results of the ISSA−VMD decomposition of Data 3. In order to know which component contains more information about the features that represent the interior of the signal, we calculated the fuzzy entropy of each component. In Figure 10b, the component with the highest fuzzy entropy after ISSA-VMD decomposition is IMF_3. Figure 10c shows the results of ICEEMDAN decomposition for IMF_3. Figure 11 shows the prediction results of Data 3.

As can be seen from Table 4, the values of RMSE are 7.188 × 10^−3^ for Informer, 7.041 × 10^−3^ for VMD−Informer, 8.436 × 10^−3^ for ICEEMDAN−Informer, 7.397 × 10^−3^ for VMD−ICEEMDAN−LSTM, 6.731 × 10^−3^ for VMD−ICEEMDAN-GRU, 8.246 × 10^−3^ for VMD−ICEEMDAN−CNN and 4.643 × 10^−3^ for VIInformer. Accordingly, compared to other methods, the method in this paper improves 2.545 × 10^−3^, 2.398 × 10^−3^, 3.973 × 10^−3^, 2.754 × 10^−3^, 2.088 × 10^−3^, 3.603 × 10^−3^, respectively. Compared with other models, VIInformer exhibits good model generalization ability. The reason is that quadratic decomposition can decompose complex a non-periodic time series into a frequency-dominated periodic sequence. The powerful feature extraction capabilities of Informer enable the decomposition prediction model to achieve optimal performance.

#### 4.3.4. *Y*-Axis Vibration Data from Vibration Sensor (Data 4)

Data 4 are the equipment *Y*-axis vibration data recorded by the vibration sensor. Figure 12a shows the result of ISSA−VMD decomposition for Data 4. As shown in Figure 12b, the component with the highest fuzzy entropy after ISSA−VMD decomposition is IMF_5. Therefore, the ICEEMDAN decomposition of IMF_5 is required, and the results are shown in Figure 12c. The final prediction results with VIInformer are shown in Figure 13. It can be seen from the forecasting diagrams that although the predicted values of the VIInformer are lower than the true values, the VIInformer can predict the trend of the data correctly. Not only that, the VIInformer can forecast the extreme values correctly to the maximum extent.

As can be seen from Table 5, all the values of MSE, MAE and RMSE of the method VIInformer are the smallest. Although the predicted value of VIInformer is lower than the true value, it predicts the trend of the sequence accurately. The results benefit from the feature extraction ability of Informer and the denoising ability of the decomposition model.

## 5. Discussion

From the prediction results of the four experiments mentioned above, it is observed that the vibration trend time series prediction based on VIInformer can better fit the real data. It can be seen that the predicted values can better fit the true values in Data 1–3. Although the predicted value in Data 4 was smaller than the true value, the model can forecast the trend of the data series better and some of the extreme values. This situation can be caused by the anisotropy of feature dimensions’ prediction capacity. It may become a new improvement point in future research. We will address the observed discrepancies for future research, such as exploring alternative modeling approaches, refining the model architecture, or incorporating additional features that better capture the complexities of the vibration signals. Compared with the results of Data 1, Data 2 and Data 4, the overwhelming performance of Data 3 is increased, and such phenomena can be caused by the compatibility between the model and the data.

Meanwhile, through ablation experiments, it is found that the use of VMD alone is also beneficial for prediction accuracy. This means that although the vibration sequence has strong randomness and volatility, there are still traces to follow, and the signal can effectively identify hidden patterns in time series data. By decomposing and predicting the original sequence and recombining the predicted data, noise reduction can be achieved to a certain extent on the original sequence. However, the prediction results after ICEEMDAN decomposition are not as good as the direct prediction results. This may be due to the excessive false components generated by ICEEMDAN decomposition. By comparing the experimental errors of four data, it can be seen that the prediction effect is best after secondary decomposition. This indicates that using fuzzy entropy for secondary decomposition and removing false components has a positive impact on the prediction results. The main reason is that the decomposition method decomposes complex non-periodic time series into a frequency-dominated periodic sequence, and the model has a strong generalization ability for such sequence data. At the same time, the classic deep learning models LSTM, GRU and CNN are selected as benchmark models combined with signal decomposition algorithms. It can be found that the Informer combined with signal decomposition performs better in predicting than the other three deep models.

## 6. Conclusions

This paper proposes a device operation trend prediction model based on VIInformer. Compared to existing methods, this model has the following advantages:(1)The randomness and volatility of equipment operation trend data after VMD decomposition decreased. The ICEEMDAN secondary decomposition of components with excessive fuzzy entropy can obtain more characteristic information containing equipment operation trends, and the non-stationary nature of the sequence after the secondary decomposition is greatly reduced.(2)Using the Informer model to predict the decomposed sequence is different from traditional machine learning models. Informer’s parallel computing ability has been significantly improved, and the model’s running speed has been effectively improved.

In summary, compared to other models, the model proposed in this article has higher prediction accuracy, more accurate and reliable prediction of equipment operation trends, and broad application prospects.

Meanwhile, there are also some areas for improvement in this study. For example, this study only used historical equipment vibration time series to conduct the prediction task. It makes sense to research the multi-temporal scale features in order to achieve more accurate prediction results. At the same time, how to improve the model runtime effectively is also a promising research direction. In addition, we were only able to include four datasets in our study due to resource constraints. This limited sample size may affect the generalizability of our findings.

In the future, we will conduct studies and research concerning time series forecasting methods. Deeper research on data with non-stationary, non-periodic fluctuations and high noise will be carried out and the impact of this problem on the forecasting operation will be solved. More self-testing data will be added in this experiment to further improve the persuasiveness of the model. Saving computational costs is valuable for practical applications. Equipment fault diagnosis prediction will be taken as the next direction of research.

## Figures and Tables

**Figure 1 sensors-23-05819-f001:**
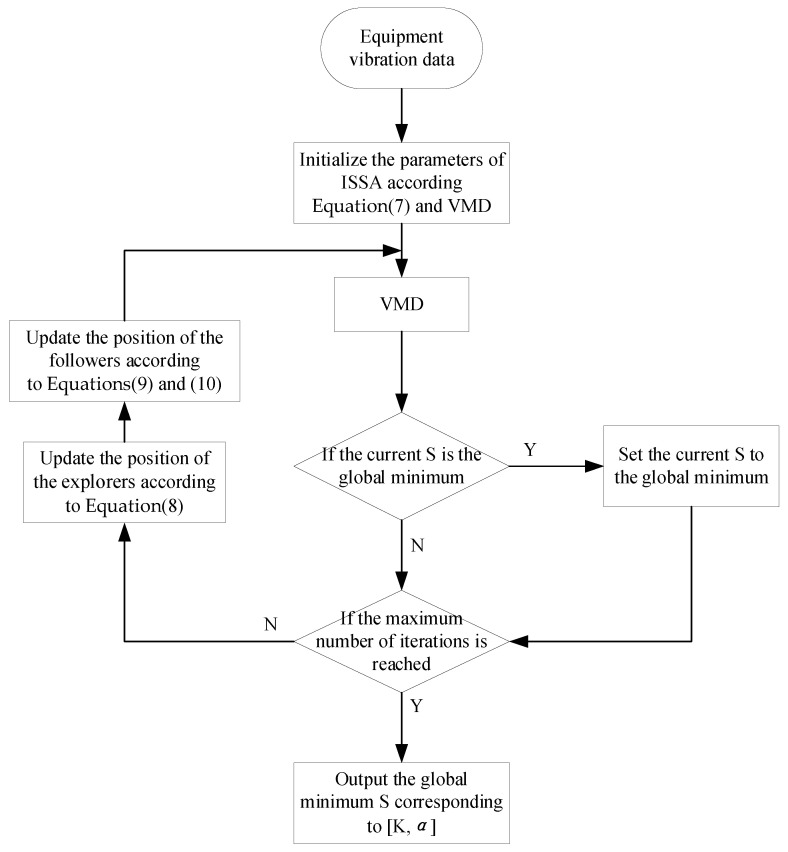
Flowchart of ISSA−VMD. The ISSA−VMD is a method of optimizing VMD using the ISSA algorithm.

**Figure 2 sensors-23-05819-f002:**
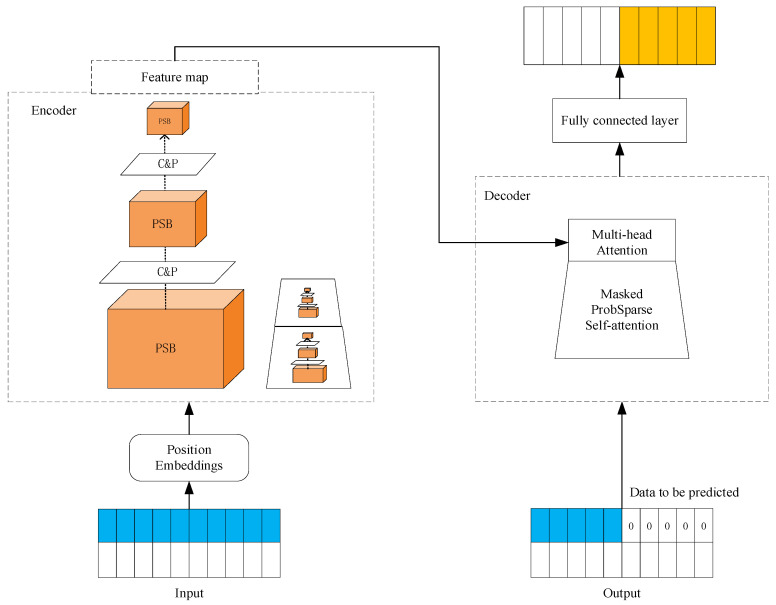
Unit structure of Informer.

**Figure 3 sensors-23-05819-f003:**
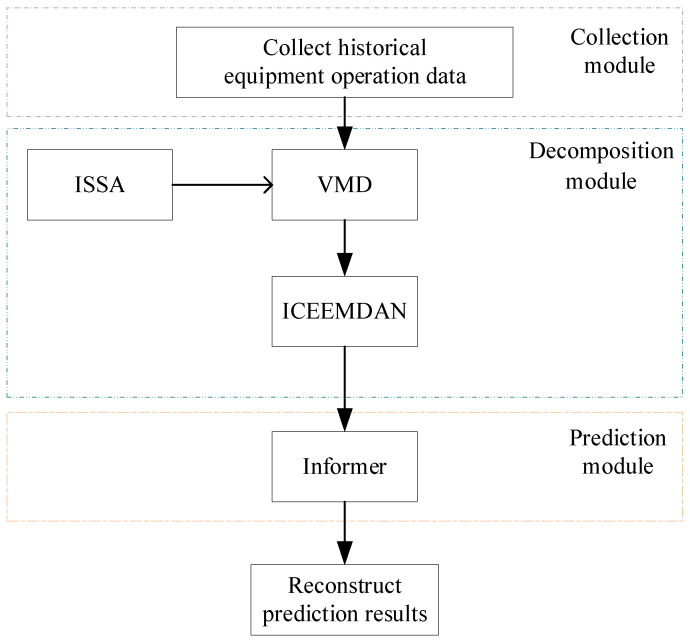
The prediction method of operation trend.

**Figure 4 sensors-23-05819-f004:**
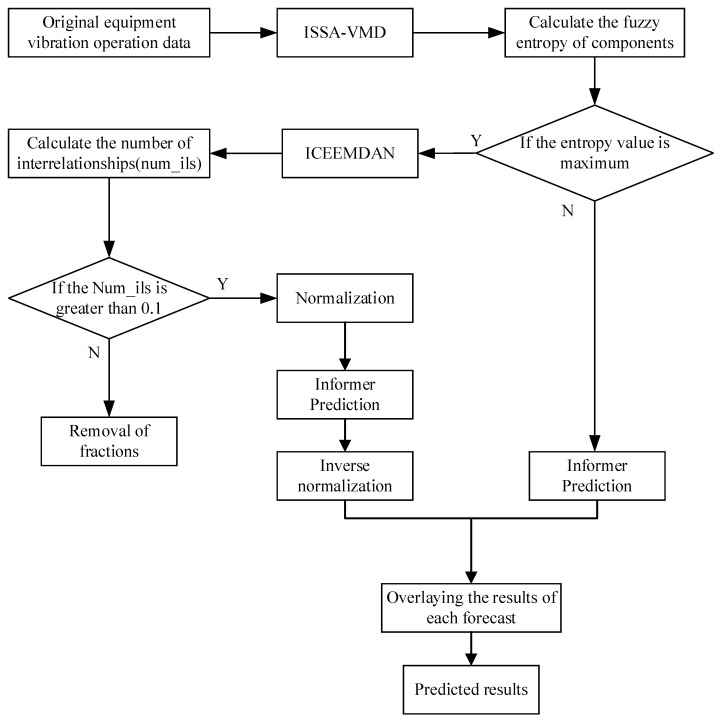
Flowchart of VIInformer.

**Figure 5 sensors-23-05819-f005:**
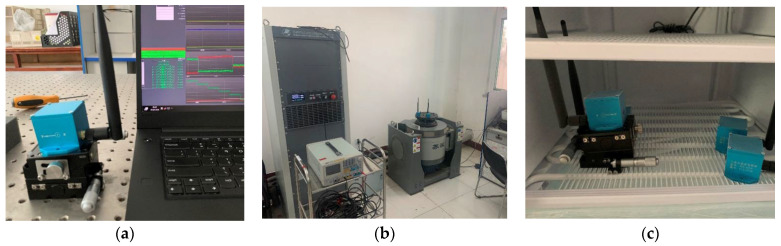
The process of Sensor testing. (**a**) The equilibrium testing of inclination sensors. (**b**) The vibration table testing of vibration sensors (**c**) Long term low-temperature box stability test.

**Figure 6 sensors-23-05819-f006:**
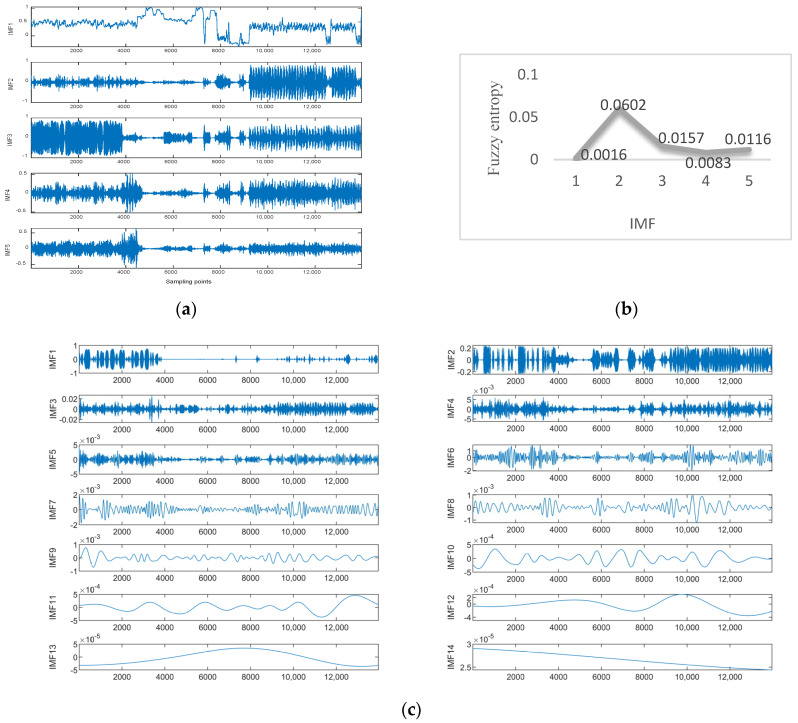
Experimental results with Signal decomposition. (**a**) ISSA−VMD decomposition results for Data 1. (**b**) Fuzzy entropy of components. (**c**) ICEEMDAN secondary decomposition results for IMF_2.

**Figure 7 sensors-23-05819-f007:**
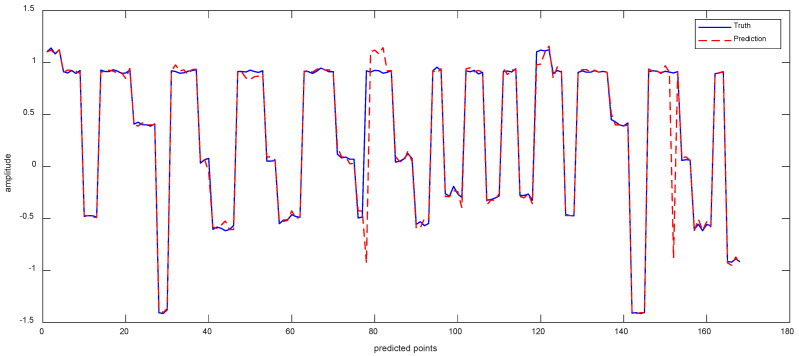
The Prediction Result of Data 1 with VIInformer.

**Figure 8 sensors-23-05819-f008:**
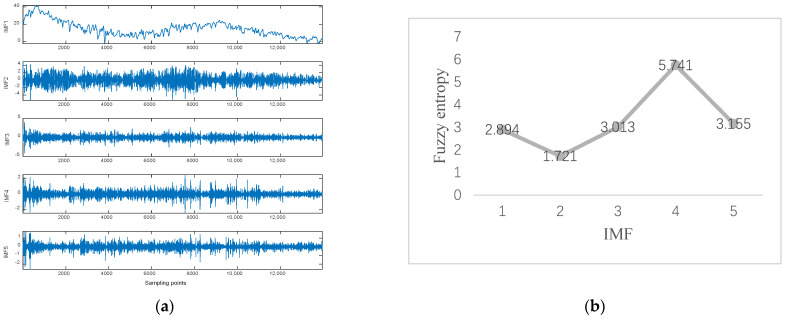
Experimental results with Signal decomposition. (**a**) ISSA−VMD decomposition results for data 2. (**b**) Fuzzy entropy of components. (**c**) ICEEMDAN secondary decomposition results for IMF_4.

**Figure 9 sensors-23-05819-f009:**
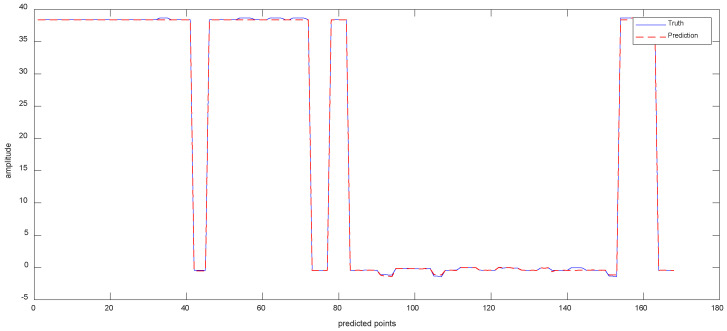
The Prediction Result of Data 2 with VIInformer.

**Figure 10 sensors-23-05819-f010:**
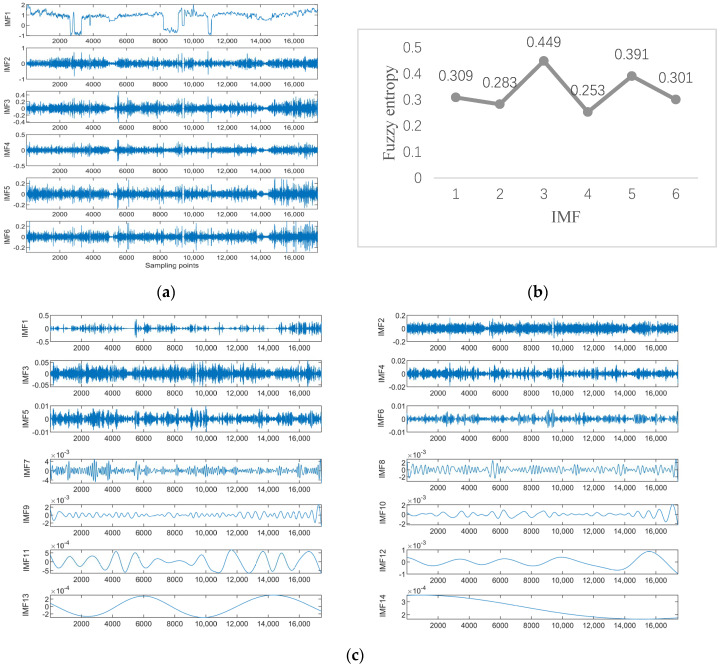
Experimental results with Signal decomposition. (**a**) ISSA−VMD decomposition results for Data 3. (**b**) Fuzzy entropy of components. (**c**) ICEEMDAN secondary decomposition results for IMF_3.

**Figure 11 sensors-23-05819-f011:**
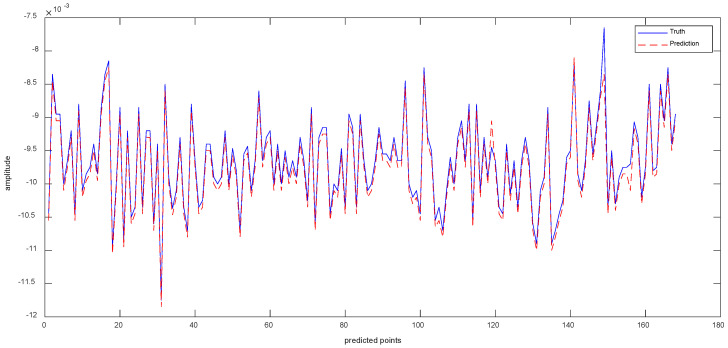
The Prediction Result of Data 3 with VIInformer.

**Figure 12 sensors-23-05819-f012:**
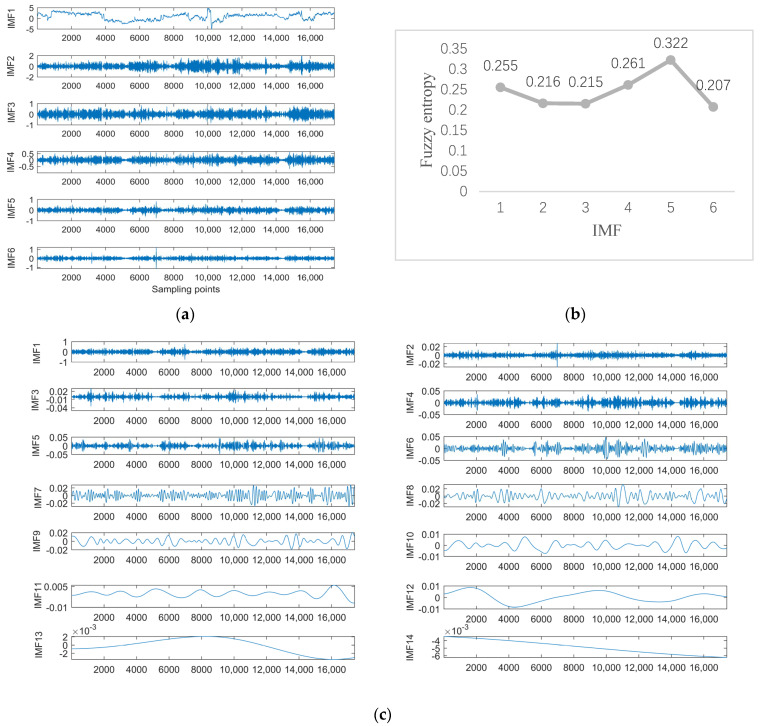
Experimental results with Signal decomposition. (**a**) ISSA−VMD decomposition results for data 4. (**b**) Fuzzy entropy of components. (**c**) ICEEMDAN secondary decomposition results for IMF_5.

**Figure 13 sensors-23-05819-f013:**
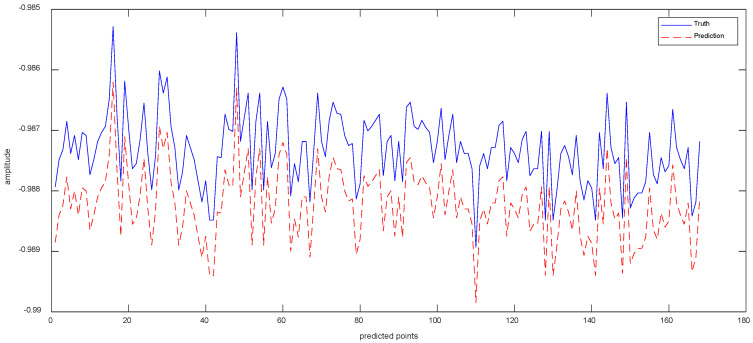
The Prediction Result of Data 4 with VIInformer.

**Table 1 sensors-23-05819-t001:** The parameters of Informer.

Parameter	Value
Encoder layers	2
ProbSparse Attention	8
Decoder layers	1
Multi-head Attention	2
Learning rate	0.00001
Batch size	32
Maximum iterations	200
Dropout	0.1
Training data length	720
Predicted length	168

**Table 2 sensors-23-05819-t002:** Comparison of errors of Data 1.

Method	MSE	MAE	RMSE	Running Time (s)
Informer	0.1027	0.1098	0.1108	258
VMD−Informer	0.0733	0.0749	0.0951	243
ICEEMEDAN−Informer	0.1895	0.1918	0.2136	364
VMD−ICEEMDAN−LSTM	0.1907	0.1949	0.2278	598
VMD−ICEEMDAN−GRU	0.1854	0.1983	0.2262	525
VMD−ICEEMDAN−CNN	0.2752	0.3274	0.3515	794
VIInformer	0.0115	0.0262	0.0334	332

**Table 3 sensors-23-05819-t003:** Comparison of errors of data 2.

Method	MSE	MAE	RMSE	Running Time (s)
Informer	0.0925	0.0992	0.0988	262
VMD−Informer	0.0733	0.0879	0.0851	251
ICEEMEDAN−Informer	0.1115	0.1368	0.1416	375
VMD−ICEEMDAN−LSTM	0.0827	0.0984	0.0987	608
VMD−ICEEMDAN−GRU	0.0833	0.0897	0.0751	561
VMD−ICEEMDAN−CNN	0.1145	0.1318	0.1036	821
VIInformer	0.0532	0.0655	0.0634	344

**Table 4 sensors-23-05819-t004:** Comparison of errors of Data 3.

Method	MSE	MAE	RMSE	Running Time (s)
Informer	6.271 × 10^−5^	6.645 × 10^−3^	7.188 × 10^−3^	288
VMD−Informer	6.303 × 10^−5^	6.214 × 10^−3^	7.041 × 10^−3^	269
ICEEMEDAN−Informer	9.805 × 10^−5^	7.804 × 10^−3^	8.436 × 10^−3^	385
VMD−ICEEMDAN−LSTM	4.783 × 10^−5^	5.949 × 10^−3^	7.397 × 10^−3^	614
VMD−ICEEMDAN−GRU	4.234 × 10^−5^	6.083 × 10^−3^	6.731 × 10^−3^	557
VMD−ICEEMDAN−CNN	6.395 × 10^−5^	7.138 × 10^−3^	8.246 × 10^−3^	847
VIInformer	2.807 × 10^−5^	3.054 × 10^−3^	4.643 × 10^−3^	371

**Table 5 sensors-23-05819-t005:** Comparison of errors of Data 4.

Method	MSE	MAE	RMSE	Running Time (s)
Informer	0.0837	0.1004	0.1058	289
VMD−Informer	0.0639	0.0984	0.0901	260
ICEEMEDAN−Informer	0.1255	0.1274	0.1316	391
VMD−ICEEMDAN−LSTM	0.0976	0.0989	0.0987	599
VMD−ICEEMDAN−GRU	0.0933	0.0983	0.0971	521
VMD−ICEEMDAN−CNN	0.1395	0.1438	0.1446	834
VIInformer	0.00095	0.00262	0.00234	368

## Data Availability

The data are unavailable due to privacy security.

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
