# Peer review of "A Deep Learning Model with Signal Decomposition and Informer Network for Equipment Vibration Trend Prediction"

_sensors, 2023, doi:10.3390/s23135819_

Round 1
Reviewer 1 Report
This paper proposed a method for predicting equipment operation trends based on the combination of signal decomposition and Informer prediction model. The author can improve the work by considering addressing the following issues:
1、 In fact, there are many algorithms for data prediction. Why are VMD-ICEEMDNA-LSTM, VMD-ICEEMDAN-GRU, VMD-ICEEMDNA-CNN, and VIInformer used in this study?
2、 How is noise considered in the data? How is the noise level of raw data evaluated?
3、 How was the data collected? Are there any test photos?
4、 Data 1 and Data 2 are of the same type, so it is not necessary to study both.
5、 The diamond-shaped box in Figure 4 can be enlarged to include all the text; the lines and data in Figure 5(b) overlap, making the numbers unclear; the fourth figure on the left column in Figure 5(c) is incompletely displayed; Some figures in Figure 7(c) are incompletely displayed.
6、 The formulas and characters in the paper need to be re-edited. The references should be clearly checked and avoid mistakes.
Moderate editing of English language required.
Author Response
Dear reviewer:
Thank you for your valuable feedback on our manuscript.
Please see the attachment.

Reviewer 2 Report
The work presents an interesting method for extracting signal characteristics using Variational Mode Decomposition for Equipment Operation Prediction. The proposed approach is interesting and provides a significant contribution to the field. I only have the following concerns:
In Section 4.2.1, the authors mention that they used an initial population of 20 and a maximum of 100 iterations in the VMD algorithm. However, no solid argument is provided regarding the choice of these values. It would be beneficial for readers to understand the rationale behind the selection of these parameters. I recommend that the authors include a brief discussion explaining how they arrived at these values and how they affect the results.
· Additionally, in the discussion of Figure 12, the authors simply state that "the final prediction results with VI Informer are shown in Figure 12." However, I believe it is necessary to delve deeper into the discussion of this figure. It would be helpful for readers if the authors explain the specific patterns or trends observed in the figure and how they support the results and conclusions presented in the article.
· In the conclusions, the authors claim that their proposed model has higher prediction accuracy compared to other models and can provide more accurate and reliable predictions of equipment operation trends, with broad application prospects. However, to substantiate this claim, it would be necessary to analyze more than the four datasets used in the study.
· Furthermore, it would be interesting to know the time required to perform the decomposition and prediction using the proposed approach. This would provide valuable insights into the efficiency of the VMD method compared to other existing approaches.
· Lastly, I suggest that the authors consider the possibility of validating the decomposition and prediction performed in this study using a different dataset from the same sensor but on a different day. This would help evaluate the generalization ability of the proposed model and demonstrate its robustness.
Overall, the work conducted is valuable and presents an interesting approach to equipment operation prediction using VMD. However, further clarity and justification are needed in certain aspects, as well as additional analysis to support the presented conclusions.
Author Response

(The authors gave the same response as above.)

Reviewer 3 Report
This study develops a method to conduct equipment operation trend prediction by combining signal decomposition and informer prediction model. The topic is interesting and suitable for this journal. However, this manuscript needs to be further improved in its novelty and academic contribution before accepting it for publication. My comments are as follows:
1. The headings of 4.3.1-4.3.4 are too general. The first sentences in these subsections should be used for better headings. Please revise them, e.g. X-axis vibration data from inclination sensor (Data 1).
2. The paper presented many results. However, the analysis only focuses on calculating the result difference in different models without any further elaboration on how different factors cause the difference.
3. The paper lacks a discussion section to compare the prediction results among the 4 datasets. It is important to illustrate how the developed model can support different signals.
4. Figure 12 shows that the prediction underestimates the vibration. However, the prediction performance well in data1-3, especially data 3. Please explain it in the discussion section
5. The limitations and possible future works of this study should be presented in an organized manner.
Author Response
Dear reviewer:
We would like to express our sincere gratitude for reviewing our research paper and providing valuable comments and suggestions.
Please see the attachment.

Round 2
Reviewer 1 Report
The authors have made a comprehensive revision according to the suggestions.
Reviewer 3 Report
The authors have successfully considered the suggestions of the reviewers. The manuscript should be considered for publication in the journal.